# Iteration Head:
# A Mechanistic Study of Chain-of-Thought

**Vivien Cabannes**
FAIR, Meta AI

**Charles Arnal**
Datashape, INRIA

**Wassim Bouaziz**
FAIR, Meta AI

**Alice Yang**
FAIR, Meta AI

**Francois Charton**
FAIR, Meta AI

**Julia Kempe**
Courant University and Center for Data Science, NYU & FAIR, Meta AI

## Abstract

Chain-of-Thought (CoT) reasoning is known to improve Large Language Models both empirically and in terms of theoretical approximation power. However, our understanding of the inner workings and conditions of apparition of CoT capabilities remains limited. This paper helps fill this gap by demonstrating how CoT reasoning emerges in transformers in a controlled and interpretable setting. In particular, we observe the appearance of a specialized attention mechanism dedicated to iterative reasoning, which we coined "iteration heads". We track both the emergence and the precise working of these iteration heads down to the attention level, and measure the transferability of the CoT skills to which they give rise between tasks.

## 1 Introduction

In the rapidly evolving field of artificial intelligence, Large Language Models (LLMs) have emerged as a pivotal component [45]. Their ability to understand, generate, and manipulate human language has opened up new avenues towards advanced machine intelligence. Interestingly, despite being primarily trained on next-token prediction tasks, LLMs are able to produce much more sophisticated answers when asked to generate steps of reasoning [30, 58]. This phenomenon, often referred to as Chain-of-Thought (CoT) reasoning, and illustrated on Table 1, appears paradoxical: on the one hand, LLMs are not explicitly programmed to reason; on the other hand, they are capable of following logical chains of thoughts to produce relatively complex answers.

**Table 1:** Chain-of-Thought consists in eliciting reasoning steps before answering (A) a question (Q).

| | | |
|---|---|---|
| [Q] What is $8 \times 8 \times 3$? | : | [A] 210. |
| [Q] What is $8 \times 8 \times 3$? Take it step by step. | : | [A] $8 \times 8 = 64$, $64 \times 3 = 192$. It is 192. |

Recent studies have shown that the class of problems a transformer can solve with single-token prediction, i.e. by outputting a single token meant to be the correct answer, is rather limited [24, 54, 15]. In contrast, when transformers are allowed to freely generate tokens before providing a final answer, they can use those generated tokens as a tape to emulate a form of Turing machine [48]. This enables them to solve a larger class of problems [38, 18, 39, 35]. However, our understanding of why and how transformers gain CoT abilities when trained with next-token predictions remains limited. We aim to provide insights on the matter.

**Summary of Contributions.** We adopt a "mechanistic interpretability" approach [see 17]: we work with simple, controlled problems and architectures that capture the key aspects of the problem and allow us to observe and analyze, down to the network's weights and attention, the emergence of CoT in our models. In practice:

38th Conference on Neural Information Processing Systems (NeurIPS 2024).

- We describe the simple yet rich setting of iterative tasks and iterative algorithms, including three simple examples: a copying, a polynomial iteration, and the parity problems.
- We explain why such problems are hard to solve for transformers with single-token prediction. Conversely, we describe how a certain distribution of weights within the first two attention layers of a transformer, which we call an "iteration head", enables a transformer to solve iterative tasks with CoT reasoning with relative ease.
- We hypothesize that iteration heads naturally appear in transformers trained on (hard enough) iterative tasks, and verify this hypothesis in small-scale experiments.
- Ablation studies demonstrate the impact of the training set and choice of hyperparameters in their emergence. We also observe the good transferability of the iterative reasoning skills granted by the attention heads from one iterative task to another, from which we deduce the usefulness of data curation.

Our controlled yet illustrative experimental setup sheds light on the emergence of CoT capabilities in larger LLMs, whose attention patterns are much harder to interpret. In particular, our experiments suggest that transformers are likely to develop "inner circuits" specially dedicated to multistep reasoning, which can then be applied, in combination with more specialized skills, to a variety of tasks that share the same underlying logical structure. This gives a credible explanation of the strong CoT reasoning capabilities of current state-of-the-art LLMs, as their training corpora (human-written texts, computer code) include many examples of complex multistep reasoning.

**Related Work.** This work is set in the realm of mechanistic interpretability [e.g., 43, 7]. A top-down line of work is trying to explicit algorithms implemented by transformers in the wild [e.g. 57, 20, 25], although some findings might be fallacious [9]. A bottom-up line of work, to which we belong, consists in building understandings from small models that are relevant for bigger models, in particular regarding in-context learning [see, e.g. 61, 19, 8, 23, 2, 34, 49, 16, 60]. In-context learning relates to the reproduction of reasoning patterns that appear in a prompt or context [10]. In contrast, our study of CoT relates to reproducing reasoning patterns that appear in the training set.

## 2   Controlled Setup: Learning Iterative Algorithms

Human language and human reasoning are often organized in a multistep, cumulative fashion, with each new thought or group of sentences building upon the ones that precede to work towards some final conclusion. LLMs naturally benefit from learning such reasoning patterns: not only are they prevalent through much of their training data, but they also represent an efficient way to divide the total processing effort required into easier intermediate steps. In what follows, *we choose to focus on iterative algorithms and iterative tasks as a controlled proxy for more general forms of CoT reasoning*. Indeed, though conceptually simple, iterative algorithms exhibit a key property: they are simultaneously hard to learn for transformers using next-token predictions, and comparatively easy to learn using CoT reasoning. As such, iterative tasks are ideally suited to illustrate the usefulness of CoT reasoning, and to study its emergence.

---
**Algorithm 1** Iterative Schemes
```
s = Init
for x in Sequence do
    s ← F(s, x);
end for
return s
```
---

We define *iterative algorithms*, or iterative schemes, as follows: an iterative algorithm is the combination of an input sequence, denoted as Sequence, and made of $L$ elements $(x_t)_{t \in [L]}$ (with $[L] = \{1, \cdots, L\}$), and an internal state, denoted as $s$, initialized to some default value $s_0 = \text{Init}$, and updated as the sequence is processed according to some rule $s_t = F(s_{t-1}, x_t)$ for some function $F$. Pseudo-code illustrating the concept is provided by Algorithm 1, see also Figure 1. By extension, we informally call iterative task a task which is naturally solved by outputting the end product of some iterative algorithm applied to some input sequence. As an example, consider the *parity problem*, i.e. the problem of computing the parity of the sum of a sequence of 0s and 1s: it can be easily framed as an iterative task. Using the notations of Algorithm 1, let the initial state Init be equal to 0, and let $F(s, x)$ be equal to 0 if $s$ is equal to $x$, and 1 otherwise. Then the final $s_L$ gives the parity of the sum. Although this task could also be solved in a non-iterative fashion, the iterative solution can be seen as simpler and more parsimonious.

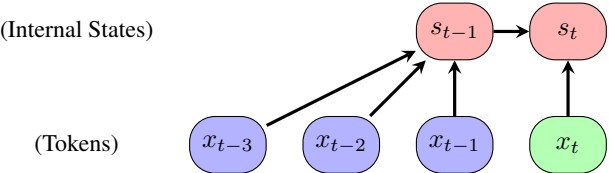

**Figure 1:** Arguably, reasoning involves updating an internal state (red) as new information is processed (green). The diagram above, where each element represents a piece of information, is an abstract depiction of this idea. This observation motivates our use of iterative tasks as a proxy for more general reasoning processes. At first glance, a limitation of transformers is their lack of an internal state, which makes it challenging to implement this diagram [32].

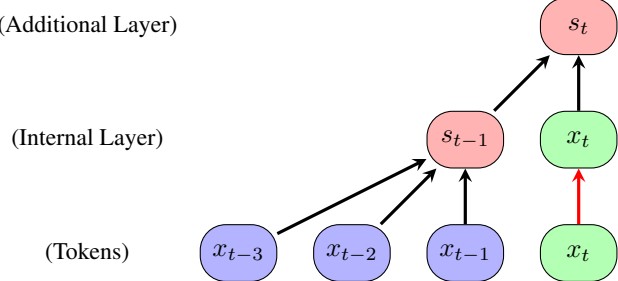

**Figure 2:** A single transformer layer cannot implement the diagram from 1, as it cannot access its previous outputs. This limitation can be bypassed by stacking transformer layers, as illustrated here. The red arrow indicates a residual connection. This naive method requires as many layers as there are reasoning hops.

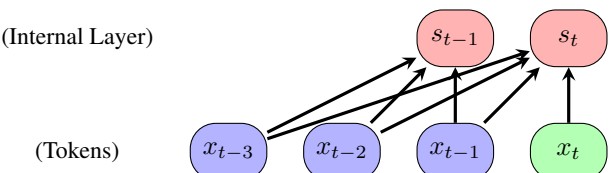

**Figure 3:** Alternatively, a transformer could compute each state $s_t$ from scratch. This implementation does not require additional layers, but it is not parsimonious, which could lead to computational inefficiencies. This explains the difficulty for a transformer to output the final answer of a chain of reasoning within a token (i.e., with next-token prediction, and without chain-of-thought)

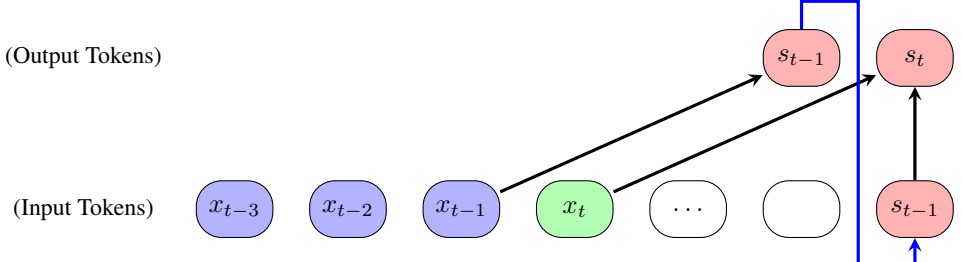

**Figure 4:** Chain-of-thought addresses this issue by explicitly representing the reasoning process in token space. The auto-regressive nature of LLMs (blue arrow) allows for the implementation of iterative algorithms, as long as the states are encoded in token space. A concrete implementation of such a mechanism, which we call *iteration head*, is described in Section 3. In practical applications of LLMs, one could imagine earlier layers summarizing the $t$-th input sentence (or some other coherent semantic information of varying token length) into $x_t$, as well as summarizing the generated CoT sentences into some $s_{t-1}$, with later layers translating the state $s_t$ into readable text.

**Can a Transformer Learn Iterative Algorithms?** Briefly summarized,[1] a transformer is composed of a series of transformer blocks and operates on the space of sequences. A transformer block performs cross-operations that combine elements of a sequence through the use of *attention heads* to generate new sequences, and parallel operations applied to each element of a sequence separately through the use of *feedforward layers* (or MLP, i.e., multi-layer perceptrons). Auto-regressive transformers in particular are trained to perform next-token prediction; in other words, from a training corpus that contains sequences $(z_t)$ of tokens, the transformer is trained to output $z_{t+1}$ from the truncated sequence $S_t = (z_r)_{r \in [t]}$.

Given a certain number of transformer blocks, a transformer can only apply a corresponding number of cross-operations to predict the next token. This limits its ability to learn even relatively simple iterative tasks, see Figures 2 and 3. E.g., consider the task where the input sequence is $(x_1, \ldots, x_L)$, possibly restricted to $x_i \in [a, b]$ for some $a < b$, and the desired output is the product $\prod x_i$. The product is multilinear in the entries of the input sequence $(x_1, \ldots, x_L)$. If we model the output of an attention layer as sums of monomials of degree at most three in its input variables (due to key-query-value interaction), this makes learning the task quite hard for a transformer, and bounds the maximum length of the sequences that a transformer with a given number of blocks can correctly process [see 51, and related literature for formal discussions on the matter that capture this log-depth dependency].

However, when transformers are allowed to generate many tokens before providing an answer, which implicitly lifts the constraint on the number of operations performed by the transformer (see Figure 4), the picture changes [48, 39, 35] [see also 14, 21]. In particular, Figure 5, explained in the next subsection, illustrates how a two-layer transformer can implement what we named an "iteration head". This potentially enables it to learn any iterative algorithm, assuming that its second layer MLP is big enough to implement any successor function $F : (x_t, s_{t-1}) \mapsto s_t$.

**Synthetic Data.** To study the emergence of CoT in controlled settings, we introduce two simple iterative problems. The first problem is a straightforward instance of Algorithm 1, where the tokens and the states are elements of the finite field $\mathbb{F}_p = \mathbb{Z}/p\mathbb{Z}$ (for some prime number $p$), i.e., integers modulo $p$, and the iterative step is the evaluation of a polynomial function $P \in \mathbb{F}_p[X, Y]$ in those two variables:

$$x \in \mathbb{F}_p, \quad \text{Init} = 0, \qquad F(s, x) = P(s, x) \qquad \text{(Polynomial Iteration)}$$

Letting $P(s, x) = s + x$ and $p = 2$, the problem reduces to the so-called parity problem:

$$x \in \{0, 1\}, \quad \text{Init} = 0, \qquad F(s, x) = s + x. \qquad \text{(Parity Problem)}$$

For ease of study, we also consider an even simpler problem: the copying problem, where the goal is simply to output an exact copy of the input sequence.

$$x \in \{0, 1\}, \qquad F(s, x) = x \qquad \text{(Binary Copy)}$$

Note that there is a small abuse of notation here, since we are interested in the unrolled sequence of states produced iteratively by Algorithm 1, rather than the last token only. While copying may seem like an overly simplistic task, it should be put in parallel with the seminal work of Olsson et al. [44] that advocates studying a copying mechanism to better understand in-context learning.

For each of our problems, we encode the data, i.e. the sequences $(z_t)$, in the following form:

$$[\texttt{Problem}] \quad [x_1] \quad [x_2] \quad \cdots \quad [x_L] \quad [\texttt{EoI}] \quad [s_1] \quad [s_2] \quad \cdots \quad [s_L] \quad [\texttt{EoS}].$$

A first token indicates the problem generating the sequence (e.g., "copy", or "parity"), after which $L$ input tokens $x_t$ are provided. The end of the input is specified by an end-of-input token (EoI). Subsequent tokens encode the states $s_t$ of Algorithm 1 at each iteration, until termination, which is indicated by an end-of-sequence token (EoS).

## 3 One Head to Rule Them All

We have discussed how transformers are limited in the iterative tasks that they can efficiently solve using only next-token prediction. By contrast, we describe in this section a certain distribution of weights which, if correctly learnt, would allow a two-layer transformer to efficiently implement iterative algorithms by using chain-of-thought reasoning. After that, we perform various experiments to identify the conditions under which this theoretical circuit does appear.

---

[1]We assume that the reader is familiar with the transformer architecture [see, e.g., 56, 36, for details].

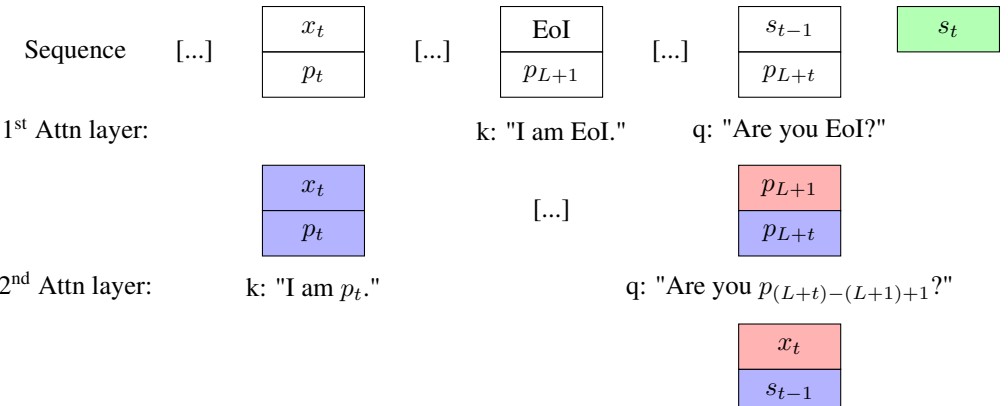

**Figure 5:** Implementation of an iteration head with a two-layer transformer. Contiguous box: superposition in high-dimensional space. Blue: information brought to working space thanks to residual connections. Red: information brought thanks to attention. Green: next-token prediction. The first layer MLP implements a subtraction $t = (L + t) - (L + 1) + 1$ for the second attention to be able to query $p_t$ from $(p_{L+1}, p_{L+t})$. The second layer MLP implements $F$ to be able to predict $s_t$ from $(s_{t-1}, x_t)$, with the "end-of-input" mark assimilated to the initial state $s_0$ of Algorithm 1.

### 3.1 Theoretical Circuit

This subsection describes a natural way to implement an iterative algorithm with a transformer. Let us consider a prompt $(x_t)_{t \in [L]}$, to which we append a special "end-of-input" (EoI) token that marks the end of the input. The completion sequence will be generated with the $t$-th new element encoding for the state variable after $t$ steps of Algorithm 1. The $t$-th new element (i.e. the $L + 1 + t$-th token of the full sequence) is produced as follows. The first attention head is tasked with retrieving the position of the end of the initial prompt, i.e. the position of the EoI token. As illustrated in Figure 5, it does so using a query-key combination which informally encodes the question "Are you EoI?" and the answer "I am EoI.". Thus it extracts the positional encoding $p_{L+1}$ (which is the value associated to the $L + 1$-th token) regardless of the sequence length $L \in \mathbb{N}$, and brings $p_{L+t}$ into its working space as well through the residual connection (we formalize such statements in the next paragraph). As shown further below in Figure 5, the next attention head then generates a query "Are you $p_t$?" from $p_{L+1}$ and $p_{L+t}$, which is answered positively by a key "I am $p_t$" associated to the $t$-th position. Hence the head retrieves the value associated to this position, which is $x_t$. It also obtains $s_{t-1}$ (or rather the approximation of it that was produced at the previous step) through the residual stream. The MLP can finally compute the new state $s_t = F(s_{t-1}, x_t)$ from $s_{t-1}$ and $x_t$. This can always be done by a large-enough MLP assuming that the second attention layer outputs all the relevant information regarding $s_{t-1}$ and $x_t$, as a result of universal approximation [27]. Note that the operations performed by the two attention layers are totally independent from the precise iterative task considered, i.e. from the choice of $F$; their only goal is to retrieve $x_t$ and $s_{t-1}$. We call the pattern of weights that realize these operations, as well as the underlying algorithm, an "iteration head".

**Information Superposition in Working Spaces.** In our description of an iteration head, we have rather informally said that some variable $x$ is "extracted" or "obtained". Formally, a transformer transforms a sequence $(x_t)_{t \in [L]}$ into a series of sequences $(e_{t,l})_{t \in [L]}$, where $l$ is an index specifying layers, and $e_{t,l} \in \mathbb{R}^d$, with $\mathbb{R}^d$ being referred to as the "working space". The input tokens and their positions are brought into working spaces using embeddings that are typically learned, then added together. Assuming that the working spaces are high-dimensional enough, and because those embeddings are learned, a transformer can use different parts of $\mathbb{R}^d$ to simultaneously store token and positional information, as if those embeddings were actually *concatenated* rather than added. Likewise, transformer layers output variables are learned functions of their input; if needed, and assuming that $d$ is large enough, $e_{t,l+1}$ can superpose some $e_{s,l}$ and $e_{r,l}$ for different $s, r \leq t$, in which case one may consider $e_{t,l+1}$ as somewhat equivalent to the concatenation of $e_{s,l}$ and $e_{r,l}$. This is why our exposition above focuses on "information pathways", i.e. which variable is generated using which variable, and sentences such as "$x_t$ and $s_{t-1}$ are brought to the working space" should be understood as "some vector encoding the relevant information of both $x_t$ and $s_{t-1}$ is produced".

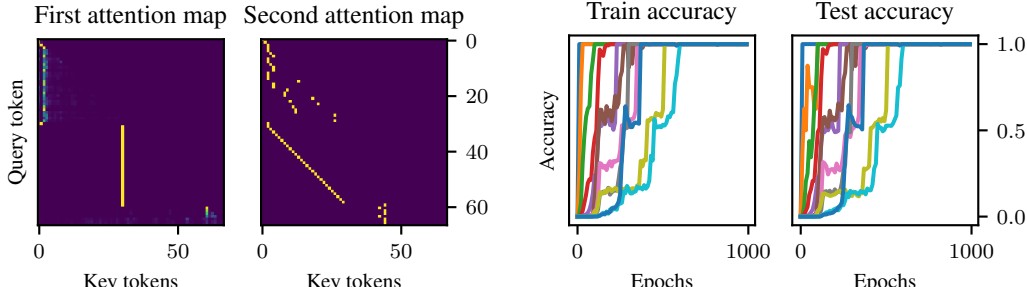

**Figure 6:** Left: attention maps learned for the parity problem when processing a sequence of length $L = 29$. Yellow indicates high attention score. The yellow line on the left plot shows that all the queries after the EoI token at position $t = 30$ point to the EoI token. In other terms, the first attention implements the "Are you EoI?" query of Figure 5, while the second implements the "Are you $p_t$?" query. Right: accuracy dynamics for different sequence lengths when learning the parity problem. We observe fast learning of short sequences (we used the `tab10` color scheme of Matplotlib [28] with $L \in \{8, 11, 14, 17, \ldots, 32\}$), and characteristic staircase behaviors.

**Approximate Iteration Heads.** Iteration heads are an efficient, flexible and parsimonious way to implement iterative algorithms; as such, we expect them to naturally emerge during training. Nonetheless, transformers have flexible architectures that can perform similar operations in different ways. Hence we also expect to see some variations with respect to the schematic architecture described above (see Figures 8 and 13), in particular when the embedding dimension becomes too small for the information superposition from the previous paragraph to be correctly implemented.

### 3.2 Learning an Iteration Head

In this subsection, we examine the circuit that a transformer actually learns when trained on an iterative task with chain-of-thought. We observe that the theoretical circuit described in the previous subsection does appear in practice.

**Experimental Design.** Unless otherwise stated, our experimental setup is as follows. Data was generated for the binary-copy, parity, and polynomial iteration problem with $P(X, Y) = XY + 1$ in $\mathbb{F}_{11}$. For each length $L$ from $L_{\min} = 1$ to $L_{\max} = 32$, we generated $n = 1024$ input sequences of length $L$ (corresponding to a total sequence length of $2L + 3$) uniformly at random for both training and testing sets, creating datasets of $N = 16,384 = 16 \times 1024$ sequences in total.[2] We utilized auto-regressive transformers [10] with two layers and one attention head per layer. The embedding dimension was set to $d = 128$, with learned absolute positional encoding added to the learned token embedding. The weights were optimized over 1000 epochs with Adam [29], a batch size of 256, and a fixed learning rate set to $\gamma = 3 \cdot 10^{-4}$, with default PyTorch parameters otherwise [46]. Our source code is available at `https://github.com/facebookresearch/pal`. Our experiments consumed 12k V100-hours.

**Attention Heads.** In our initial experiment, we trained a transformer to solve either the parity task or the copying task only. The "iteration head" pattern of weights, described in the previous sub-section, can be seen in the attention maps of the first and second attention layers: an example is reported in Figure 6. Namely, we observe that when the model produces the $(L + t + 1)$-th token (meant to be $s_t$), the following happens. The attention of the first transformer block is fully focused on the position of the EoI token, corresponding to the informal query "Is this token equal to EoI?", creating a yellow line on the left of Figure 6. This allows the first attention layer to retrieve the positional encoding $p_{L+1}$ of the EoI token, in addition to the positional encoding $p_{L+t}$ of the last token of the current sequence (the state $s_{t-1}$) coming from the residual stream. Using this information, the second attention layer is able to generate the informal query "Is this token in position $t$?", to extract some encoding of the token $x_t$. Consequently, the attention of the second layer is fully focused on the position of the $t$-th entry, creating the yellow off-diagonal line on the second plot of

---

[2]Note that, even when $x_t \in \mathbb{F}_2$, the expected percentage of collision between the training and testing set decreases exponentially fast with $L$, ensuring minimal contamination between training and testing.

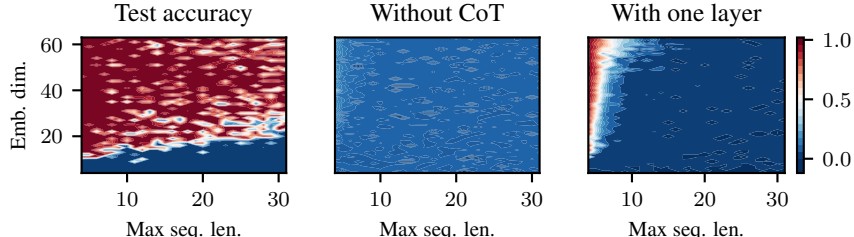

**Figure 7:** Test accuracy (where red indicates better performance) after learning the polynomial iteration task with $P(X, Y) = XY + 1$ in $\mathbb{F}_{11}$ for 1000 epochs. The accuracy is reported as a function of the embedding dimension (on the $y$-axis), and the maximum sequence length $L_{\max}$ (on the $x$-axis). The learning was conducted with a two-layer transformer with CoT (left), without CoT (middle), or with a one-layer transformer with CoT (right). This illustrates the usefulness of CoT and two-layer architectures.

Figure 6. Using the information of $x_t$ and $s_t$, the following MLP can then compute $s_t = F(x_t, s_t)$. For a given sequence length $L$, the learned attention maps were found to be invariant to the input token $(x_t)_{t \in [L]}$: the standard deviation of attention patterns computed over all the data was negligible.

**Successor Function.** We empirically verified that after training a two-layer transformer with one attention head per layer on the copying dataset, fine-tuning only the second layer MLP on parity data enabled us to achieve 100% accuracy on the parity problem. In the context described previously, this transfer was accomplished in fewer than 20 epochs of fine-tuning on the parity dataset. This confirms that the feed-forward layer of the second transformer block is computing the successor function $F$. In more general contexts, we found the successor function to be implemented jointly by the second layer MLP, the second attention values and output matrices, as well as the un-embedding matrix.

**Position Subtraction.** The accuracy of the model decreases with the embedding dimension $d$, as shown on the left of Figure 7. Figure 8 suggests that when $d$ is small, the first attention layer remains capable to accurately locate the "EoI" token, but the second attention layer struggles to retrieve $x_t$. This can be explained as follows: in a model that implements an iteration head, the first layer's MLP, in conjunction with the second attention key and query matrices, is expected to generate the query-key pair "Are you $p_t$" and "I am $p_t''$" by transforming $p_t$ on the one hand, and a superposition of $p_{L+t}$ and $p_{L+1}$ on the other hand, so that the end results are aligned. More abstractly, this encodes the positional subtraction $L + t - (L + 1) + 1 = t$. In high dimensions, it is relatively easy to find a set of weights to align a large number of vectors (viz., the ones encoding for $(p_{L+1}, p_{L+t})$ and for $p_t$). However, in lower dimensions, this can only be achieved when the vectors form certain special geometrical patterns [see e.g. 40, 63], which the model struggles to learn in our setting, at least with the optimization choices we made.

**Evaluation Dynamics.** With a sufficiently small model, we might expect to understand the training dynamics quite well, which could in turn provide insights on design choices for larger models to minimize training costs. While a detailed study of the training dynamics of our two-layer transformers is beyond the scope of this paper [see e.g., 41], we note several interesting facts that align with recent findings in the literature, such as the staircase profile of accuracy plots in Figure 6 [1, 3, 13], as well as the usefulness of small batch sizes and large learning rates reported in Figure 12 in Appendix [11] despite the risk of loss spikes [12, 59].

### 3.3 Ablation Studies

In addition to visualizing the attention map, we validated the learning of iteration heads through attention patching, i.e., intervening to "patch" certain attention maps. Specifically, we observed that patching the ideal attention maps (i.e., zeroing out other routes) does not disrupt perfect accuracy. In contrast, zeroing out the focus on the EoI by the first attention head, or on $p_t$ by the second, reduced performance to near random.

**Next-token Prediction; One or Two Layers.** As an initial ablation study, we considered the polynomial iteration problem with $P(X, Y) = XY + 1$ in $\mathbb{F}_{11}$, and compared the performance of

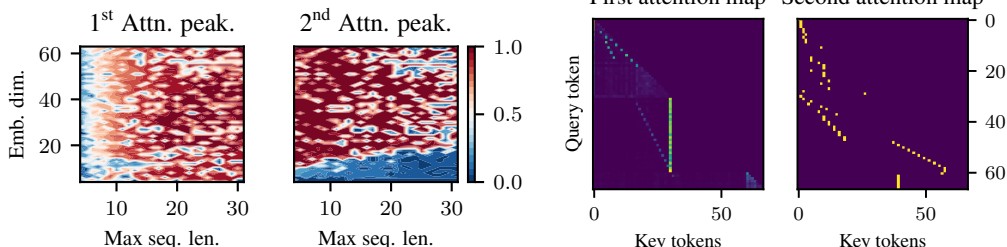

**Figure 8:** Left: attention peakiness score after 1000 epochs of learning with the polynomial iteration task parameterized by $P(X, Y) = XY + 1$ in $\mathbb{F}_{11}$ as a function of the embedding dimension $d$ and the maximum sequence length $L_{\max}$. Right: example of attention maps of sub-sampled iteration heads.

CoT reasoning with next-token prediction (i.e., without CoT), as well as CoT with a single layer transformer. Two parameters come into play: the length of the sequence, which can be seen as a difficulty parameter regarding the data; and the embedding dimension, which can be seen as a model capacity parameter [31, 55, 53]. The results, unequivocal in favor of CoT and two-layer transformers, are reported in Figure 7.

**Alternative Circuits.** Next, we explored other circuits that a two-layer transformer can learn to perform the same tasks as an iteration head. We proceed by assigning a score to measure how closely the attention maps follow the patterns of Figure 6. For the first attention map, we would like a measure of the concentration of the attention at the "Are you EoI? I am" query-key pairs, which correspond to the vertical yellow line from the left of Figure 6. For the second attention map, we would like a measure of the concentration of the attention at the "Are you $p_t$? I am" query-key pairs, which corresponds to the yellow off-diagonal from Figure 6. To avoid scaling issues, we define an attention score $a_i$ as "peaky" if it is greater than 50% after softmax averaging. We then measure the average number of peaky scores (within one sequence, and over sequences), i.e., we compute $\sum \mathbf{1}_{a_i > .5}$ instead of $\sum a_i$. This provides a clear measure of the degree to which a transformer is implementing the attention mechanisms described in the previous section.

On the left of Figure 8, we report the average peakiness found for the first and second attention layers when training a transformer for different maximum sequence lengths $L_{\max}$ and embedding dimensions $d$. We only run one experiment per pair $(L_{\max}, d)$ with a fixed random seed. The texture of the figure indicates a certain randomness between runs for similar pairs $(L_{\max}, d)$. The first attention layer almost always learns the "Are you EoI?" query-key combination, except when the maximum sequence lengths are very small. In these cases, the transformer might find different circuits to solve for different sequence lengths.

The second attention layer tends to vary more. In particular, for small embedding dimensions, the position subtraction might be challenging for the transformer to perform, leading it to find alternative mechanisms. For instance, the first layer attention might perform a previous token copy when processing the input tokens, superposing the current token $x_t$ and the previous one $x_{t-1}$ in the current working space. This allows the second layer to solely point at every other position, e.g., only attend even positions $t \in 2 \cdot \mathbb{N}$, either recovering the current token $x_t$, or the previous one $x_{t-1}$. Implementing position subtraction towards even positions only reduces the learning capacity needed by the first layer MLP [see 11, for related scaling laws]. Such a sub-sampling mechanism is notably observed on the right of Figure 8.

## 4 Skill Transfer

Some skills might be easier to acquire when trained on certain data rather than others, highlighting the importance of data curation when training LLMs. For example, datasets of code or math [e.g., 26] might exhibit formal reasoning structures that compel LLMs to learn multistep reasoning patterns when trained on them, leading to improved reasoning abilities of the final model, even in plain English [see, e.g., 37]. Our synthetic problems are ideally suited to highlight the mechanisms at play in these observations. This section illustrates how strategic data curation can facilitate learning to solve the parity problem. The crux is to find a dataset that helps the creation of iteration heads, which, once present, significantly eases the learning of the parity problem by a transformer.

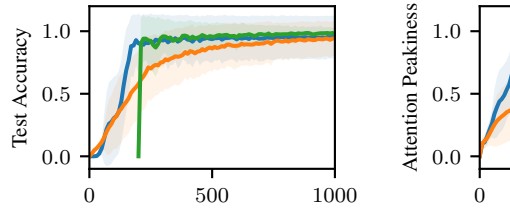 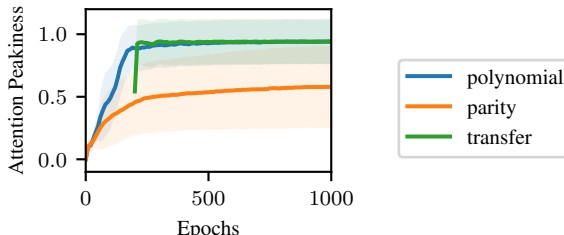

**Figure 9:** Left: Test accuracy as a function of the number of epochs, averaged over 100 runs, when learning the polynomial iteration task with $P(X, Y) = XY + 1$ in $\mathbb{F}_{11}$ (blue) and the parity problem (orange). Right: The second attention peakiness score indicates whether the network is learning the iteration head described in Figure 5. The green curve corresponds to the accuracy on the parity problem when learning the polynomial iteration for the first 200 epochs before switching the dataset to learn the parity problem.

## 4.1 Inducing Induction

We now address a simple question: can we "pretrain" a model on a task A, and then "finetune" it on a task B in order to learn to solve the task B with a smaller total number of flops than if we were to learn the task B from scratch? We will see that the answer is positive.

Figure 9 compares three learning scenarios. The learning of the polynomial iteration task corresponding to $P(X, Y) = XY + 1$ in $\mathbb{F}_{11}$ is reported in blue. The learning of the parity problem is reported in orange. Finally, the green curve represents training on the polynomial iteration task for 200 epochs (these epochs are not reported in the graph, hence the curve offset), before switching tasks and continuing the training on the parity problem. When switching from the polynomial iteration dataset to the parity dataset, we chose to reset the Adam buffers to zero. Moreover, our default experimental parameters were changed to $L_{\max} = 16$ and $n = 512$, generating training and testing sets of $N = 8,192 = 16 \times 512$ sequences. The left side of Figure 9 reports testing accuracy averaged over 100 runs, along with its standard deviation. The polynomial iteration task is learned relatively quickly, while the parity problem takes longer. The right side of Figure 9 reports the second attention peakiness score, capturing whether or not the second attention is implementing the "Are you $p_t$?" query. After 200 epochs of training with the polynomial iteration task, the iteration head is formed, and fine-tuning the network on the parity problem for less than 30 epochs enables the reuse of this circuit on the parity data (green curve, right plot), thus solving the parity task (green curve, left plot). Overall, the data curation represented by the green plot enables the computation of parities in less than 300 epochs, compared to 1000 epochs when learning solely with parity data.

This example provides a controlled setup to understand the usefulness of data curation when training larger models. It biases the model toward the implementation of specific circuits. In particular, adding code or math datasets to the training of LLMs might induce the learning of more circuits that implement various forms of reasoning patterns. These could be viewed as atomic skills that could be reused to solve more generic problems [see, e.g., 4, for further discussions on skill factorization].

## 4.2 The Role of Inductive Biases

To illustrate the usefulness of data curation and skill transfer, we needed to find a problem that is hard to learn from scratch. The parity problem was well-suited to play this role in our synthetic setting. On the other hand, the polynomial iteration task with $P(X, Y) = XY + 1$ in $\mathbb{F}_{11}$ was the easier task. One might wonder why learning with $P(X, Y) = XY + 1$ in $\mathbb{F}_{11}$ turned out to be a simpler task than learning parities, which corresponds to $P(X, Y) = X + Y$ in $\mathbb{F}_2$. Our intuition is that the parity problem can be solved in many different ways, which leads to competing signals in the gradient for updating the weights, reminiscent of the theoretical study by Shalev-Shwartz et al. [52] [see also 50, 63]. For example, we see on the right of Figure 9 that the standard deviation of the attention peakiness score is quite high when learning with parity data. This can also be observed from the texture in Figure 11 in the Appendix. This creates a *challenging optimization landscape*. In contrast, the polynomial $P(X, Y) = XY + 1$ was chosen to make the final state dependent on the token order. Removing permutation invariance is useful to reduce the variety of circuits that can solve the polynomial iteration task, and seems to speed up the training dynamics. Finally, starting from a pretrained model that already implements an iteration head creates a strong inductive bias toward the

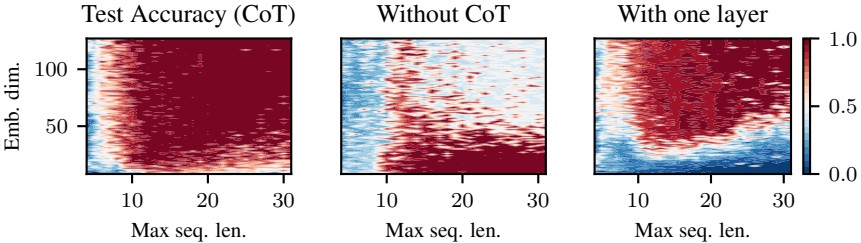

**Figure 10:** Same as Figure 7, except that we considered the parity problem, and 5000 training epochs.

iteration head circuit to solve the parity problem, allowing the parity problem to be learned within a very small number of epochs.

To deepen our understanding of the parity problem, we conducted the same scaling study depicted in Figure 7 using the parity dataset. The results are presented in Figure 10. Training was conducted over 5000 epochs. The data generation process was slightly modified to generate all sequences of length less than $L = \log_2(1048)$, and these were evenly split between training and testing (instead of generating redundant random sequences). In some sense, the parity problem can be considered relatively easy to solve in a non-iterative fashion: simply add all the elements of sequences, and reduce the sum modulo two. In theory, a single-layer transformer can use uniform attention to bring all the input tokens into superposition as input for the two-layer MLP layer, which is a universal approximator [27]. As a result, the parity with a fixed sequence length can be solved with such an architecture [see e.g. 5]. Indeed, the bottom right of the left and middle plots in Figure 10 indicate that next-token prediction performs better than chain-of-thought for sequence lengths up to $L_{\max} = 32$ with an embedding dimension of $d = 32$. This is due to the difficulty of performing position subtraction necessary for the CoT circuit, compared to the relative ease of performing addition of up to 32 bits with our two-layer architecture. Similarly, we found that a one-layer transformer was able to learn to produce correct CoT sequences for this task, demonstrating the existence of circuits fundamentally different from our iteration head to solve it. Anecdotally, the top right of the left and middle plots of Figure 10 indicate that as the model capacity increases, next-token prediction tends to overfit the training data, while CoT induces the transformer toward understanding the underlying structure that generated the data.

## 5 Conclusion

In this paper, we have explored the emergence of Chain-of-Thought (CoT) reasoning in Large Language Models (LLMs) through the lens of iterative algorithms. We have shown that, despite being trained on next-token prediction tasks, transformers can learn to solve iterative tasks efficiently using CoT reasoning. In particular, we have demonstrated that a two-layer transformer can implement what we named an "iteration head", enabling it to learn any iterative algorithm, assuming that it has enough feedforward layers following its two transformer blocks.

We have also shown that data curation can play a significant role in guiding the model towards the implementation of specific circuits. While our study has focused on simple, controlled problems and architectures, we hope that our findings shed light on the emergence of CoT capabilities in larger LLMs, whose attention patterns are much harder to interpret. In particular, they suggest that transformers are likely to develop "inner circuits" dedicated to multistep reasoning, which can then be applied to a variety of tasks that share the same underlying logical structure.

Interestingly, our work also highlights a limitation of the transformer architecture: they are stateless models. Indeed, our CoT implementation of Algorithm 1 requires the generated states $(s_t)$ to have a token representation. This allows us to recover the state of the iterative algorithm at the root (i.e., the input) of the transformer. For complex iterative algorithms, or generic language modeling, it would be more logical to maintain a state internal to the model in the embedding space. The fact that GPT architectures do not allow this is arguably a shortcoming of the current transformer architecture [see 32, 6, 22, 47, 62, for interesting discussions], [see also 42, 33].

**Acknowledgments.** The author thanks Alberto Bietti, Carles Domingo-Enrich, and Denny Wu for useful discussions.

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

**Societal Impact.** Mechanistic interpretability is focused on understanding the key mechanisms at play in deep learning systems by tracing them down to the weights. It is often associated with AI safety, hoping that a deeper understanding of these systems can help us steer them to be more "aligned" with "human values", and prevent AI dystopia scenarios. On the other hand, it could also prove useful in training more powerful models, which is associated with significant societal issues linked to the rise of advanced AI systems. These issues are too broad to be discussed in this paragraph.

# A    Additional Figures & Findings

Figure 11 studies the probability of finding the iteration head when learning with the parity data. As mentioned in the main text, it showcases the bigger probability of learning other circuits when learning the parity problem only.

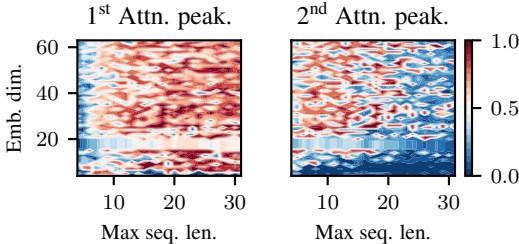

**Figure 11:** Same figure as Figure 8 yet when learning with the parity dataset. The whitening of the row around $d = 20$ is due to GPU failure, and should not be considered when parsing this figure.

Figure 12 showcases the usefulness of large learning rates and small batch size when using SGD, and the correction brought by Adam.

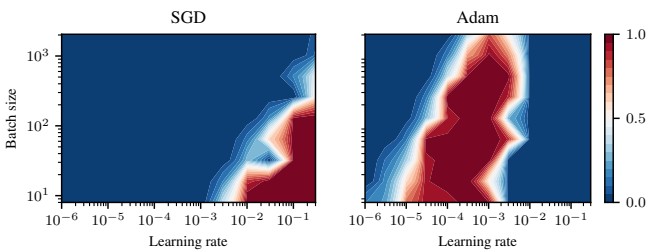

**Figure 12:** Test accuracy for SGD and Adam after 100 epochs.

Finally, Figure 13 plots some attention maps when learning with a three layers transformer with two attention heads per layer. Knowing the iteration head circuit, we are able to observe a similar circuit, yet with work shared across heads and layers.

**Position embeddings.** When we started this project, we were expecting to find some grokking structure emerged from the need to perform position subtraction. In particular, as mentionned in the main text, we were expecting this mechanism to appear as the position embedding dimension was small. When learning on the parity problem, we found that the network was not implementing the iteration head when the position embedding dimension was really small. This can notably be seen on Figures 15 and 14. We notably observed that freezing the position embedding does not change much the picture, which can be seen as a result of overparameterization. Similar type of observation were observed when learning with the polynomial iteration problem, as reported on Figure 16.

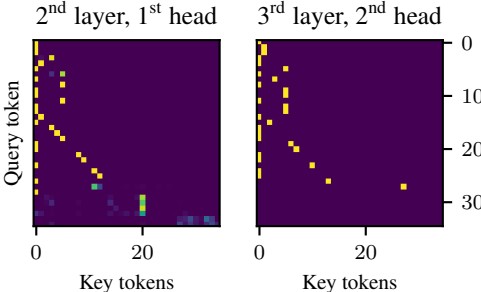

**Figure 13:** Recovering the "who is $p_t$?" key-query association, yet shared across layers and heads when training a three layers transformer with two attention heads per layer.

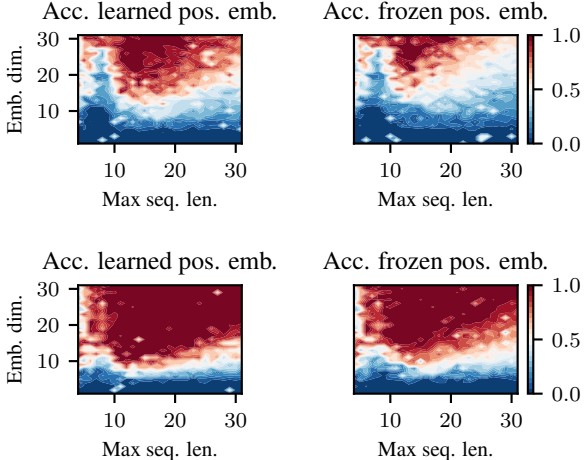

**Figure 14:** The effect of small embeddings when learning the parity problem. The top row corresponds to what has been learned after 1000 epochs. The bottom one corresponds to 5000 epochs.

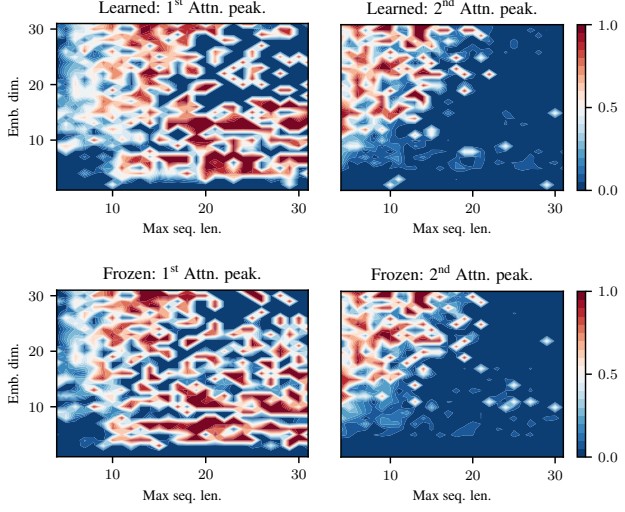

**Figure 15:** Attention learned when studying frozen vs learned positional embedding.

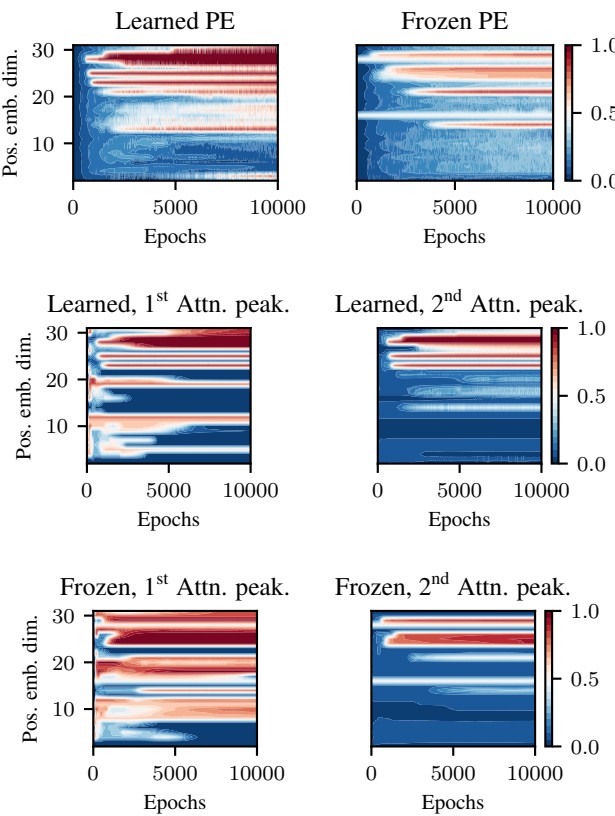

**Figure 16:** Attention learned when studying frozen vs learned positional embedding. The setting is slightly different, we fixed the token embedding dimension to 32, and added the position embedding only on the first $p$ dimension, where $p$ was varying from 2 to 32.

