# OpenReview forum: "Iteration Head: A Mechanistic Study of Chain-of-Thought"
_NeurIPS.cc/2024/Conference — NeurIPS 2024 poster_

### Official Review · Reviewer_r3gz · 2024-06-26

**Soundness:** 3
**Presentation:** 3
**Contribution:** 3
**Rating:** 7
**Confidence:** 3

**Summary:**

This paper describes an algorithm called the "iteration-head" by which autoregressive transformers can implement simple iterative functions, which may be related to the chain of thought (CoT) reasoning capabilities of language models. The paper shows empirically that simple two-layer transformers can learn to implement the iteration-head algorithm when trained on iterative tasks.

**Strengths:**

- The paper addresses a highly relevant topic: Understanding the mechanism underlying CoT capabilities of transformers.
- The paper is well-written, and studies a simple, controlled setting where results can be easily replicated.
- The empirical findings support the theoretical intuition provided. Using transfer fine-tuning to verify the role of the MLP as $F$ in the iteration head algorithm is a neat contribution.

**Weaknesses:**

- While the paper presents empirical evidence showing iteration heads can arise in small models, CoT prompting has been shown to have the strongest effects in larger models [1]. Because of this, the paper would benefit by including an analysis of whether the iteration-head algorithm is replicated in larger models, which benefit the most from CoT prompting on more complicated reasoning tasks.

- Even in settings where the tasks studied are relatively simple (polynomial iteration, binary copying and parity problem), the "iteration head" algorithm is shown to be implemented in multiple ways [Lines 251-259, Figure 4;right, and Figure 9]. The fact that the implementation can be distributed across multiple heads/layers suggests that the algorithm description provided is helpful, but incomplete. Relating how a potentially distributed implementation of the iteration head algorithm might work would strengthen the argument that this sort of computation could arise generally in pretrained transformers.

[1] Wei, et al. Chain-of-thought prompting elicits reasoning in large language models, 2023.  (https://proceedings.neurips.cc/paper_files/paper/2022/hash/9d5609613524ecf4f15af0f7b31abca4-Abstract-Conference.html)

**Questions:**

- The "iteration head" algorithm introduced and described here is very similar to the “induction head” algorithm from Elhage, et al, and Olsson et al. [2,3], with a few key differences. (1) The iteration head matches query-key vectors by position instead of matching by token/content.  (2) The iteration head algorithm also allows for more complex operations beyond literal copying by using a general function $F$ instead. A discussion contrasting the iteration head algorithm with the induction head algorithm would provide some additional clarity on why this algorithm (the iteration head) is needed, and perhaps why induction heads would not be sufficient for enabling CoT reasoning in the cases you've presented.

- Just curious -- if relative/rotary positional embeddings are used instead of absolute positional embeddings would you still need an MLP to help compute the positional difference (i.e. $p_{(L+t) - (L+1)+1}$), or could it be done solely via attention?

[2] Elhage, et al. Elhage, et al. A Mathematical Framework for Transformer Circuits. 2021 (https://transformer-circuits.pub/2021/framework/index.html)

[3] Olsson, et al. In-context learning and induction heads. Transformer Circuits Thread, 2022. (https://transformer-circuits.pub/2022/in-context-learning-and-induction-heads/index.html)

**Limitations:**

Yes. The authors are aware of the scope of their work and its limitations.

---

> ### Author Rebuttal · Authors · 2024-08-05
>
> Thank you very much for your positive review.
>
> We will integrate a longer discussion to compare our circuit with induction heads. We understand that induction heads were introduced as an example of how to move information across "registers" (the $(e_{t,l})$) with copying mechanisms. The induction head circuit is based on a copy of the previous token computed thanks to position information, together with a copying mechanism based on semantic information. The iteration head relies on another copying mechanism to move information around, in particular, the iteration copy mechanism is purely positional.
>
> We should have been clearer that the goal of our paper is not to study iterative circuits in production-size LLMs, but to introduce a controlled setting that enables rigorous study of many phenomena that are relevant for LLMs. In particular, the Llama and Mistral models use rotary embedding, while we consider absolute position embeddings (APE).
>
> We test in practice that a rotary positional embedding (RoPE) can not easily implement iteration heads with only two layers. Indeed, rotary embedding easily implements an offset attention focused on the position $t-L$ when processing the position $t$ for some fixed $L$ (no need for MLP here). Yet, iteration heads need to set $L$ to be equal to the random sequence length. To recover the length $L$, a first layer could count the number of tokens before EoI, and a second layer could be focused on EoI to recover the length $L$, before using this information to recover a head that implements the right attention "$t-L$" in a third layer. The attention maps learned in practice are reported in the rebuttal pdf.
>
> We appreciate your curiosity regarding RoPEs, and hope that our rebuttal will help you better understand and appreciate our perspective on our work.

---

> > ### Comment · Reviewer_r3gz · 2024-08-08
> > **Thank you for the reply**
> >
> > Thank you for the additional information and results regarding Transformers trained with RoPEs, I found it helpful. I also look forward to the mentioned future work idea related to “un-sharding” circuits where semantic operations may be split across layers. All of my other questions/concerns have been addressed and I think the authors’ proposed clarifications will strengthen the paper, though it is already a solid submission. As such, I have raised my score to a 7.

---

> > > ### Author Response · Authors · 2024-08-08
> > > **Thank you**
> > >
> > > Thank you very much for your quick answer, raising your score, and encouraging us to pursue a “un-sharding” study.

---

### Official Review · Reviewer_sSSN · 2024-07-07

**Soundness:** 3
**Presentation:** 2
**Contribution:** 2
**Rating:** 6
**Confidence:** 4

**Summary:**

- The authors try to understand Chain of Thought in LLMs by studying tiny toy autoregressive transformers trained on algorithmic tasks (like parity, or polynomial iteration) that are much easier to solve iteratively, and which try to be a proxy for problems LLMs solve with CoT
- These toy tasks have the form "take as input a sequence of n tokens, and output a sequence of n tokens, where the kth output token is a function of the kth input token and the k-1th output token", where the same model can handle a range of values of n.
- A crucial subtask is thus to attend from the k-1th output token to the kth input token. The authors propose a circuit to solve this consisting of an "attend to EoI" head and an "iteration head" composing in two different layers.
- They convincingly explain how this can be achieved theoretically
- They provide some evidence that it is learned in practice by showing that the predicted attention patterns are learned, but do not mechanistically show the function predicted is implemented
- The authors show that a model learns to do the parity task much faster if it is first "pretrained" on polynomial iteration, then finetuned on parity. Their interpretation is that the iteration head is a general algorithm for iterative tasks and can be repurposed for parity, but was easier to learn via polynomial iteration.

**Strengths:**

- Mechanistic interpretability of chain of thought is an important problem, and this work makes some progress
- The iteration head is an elegant algorithm for a non-trivial problem, and my best guess is that it is, in fact, learned in their setting
- The result that pretraining on polynomial iteration is a fast way to learn parity was surprising to me.
- They did many careful and thorough hyper-parameter sweeps, which resulted in interesting heatmaps

**Weaknesses:**

- The iteration head task is significantly easier than real chain of thought, making the applicability of this work to LLMs unclear. In particular, real chain of thought has varying numbers of token per step, meaning the algorithm must be much more complex than a head attending with a fixed offset per input.
- No evidence is provided that anything like iteration heads are learned in LLMs, the paper purely studies toy models
- The evidence that iteration heads are, in fact, learned in the toy model is only circumstantial - it is showed that there's some of the predicted dependence on hyper-parameters, and that the attention patterns predicted form, but alternate hypotheses for the same attention patterns are not discussed or ruled out. I would be more compelled by weights-based analysis, as in A Mathematical Framework for Transformer Circuits, or Progress Measures for Grokking via Mechanistic Interpretability.

**Questions:**

# Major Comments
- *The following are things that, if addressed, might increase my score*
- While toy models *can* be extremely enlightening if well-chosen, I think that most toy models work falls short due to failures in the model - if the toy model is a poor proxy for the task of interest, then the quality of the analysis is irrelevant. I broadly believe the author's story about iteration heads in the setting of this toy task, but their relevance to CoT in LLMs is unclear, and that is the main thing that could make this paper interesting.
- The toy tasks here are much simpler than eg a MATH problem, as each step implements exactly the same algorithm (across *all* prompts) and each step has the same offset across an input (though not across prompts) and each step only needs to output one token (rather than generating a series of tokens, before figuring out the next step). It wouldn't surprise me if real LLMs had something like a "soft" iteration head that attends somewhere in the corresponding next sentence in a MATH problem, but I don't think the paper provides much evidence for this beyond my existing priors
- I would be excited if the authors could either convince me that their toy task is a better proxy than I think it is, or show evidence that iteration head like things arise in LLMs
- Example experiment: Find some CoT questions with structured input and output, eg multi-hop factual recall or simple maths questions, and an LLM that can do them (maybe with few shot learning so it keeps to the right structure). Look for iteration-y heads which attend from the k-1th line in the output (or final token of the line) to the kth (or k-1th) line in the input (taking the sum over all tokens in that line). Try ablating each head one at a time, and look at damage to CoT performance. Show that this is anomalously high when ablating iteration-y heads. (This may have issues with hydra effects where ablating a head does little - in this case, try ablating all iteration-y heads at once)
- But I still think the paper is a weak accept rather than a reject, as it is technically solid work, iteration heads are an interesting idea and contribution that are plausibly relevant, and I found the finetuning results striking.

# Minor Comments
- *The following are unlikely to change my score, but are comments and suggestions that I hope will improve the paper, and I leave it up to the authors whether to implement them*
- I found Figure 1 very clarifying, and would recommend moving it to page 1 or 2, where it would have helped me substantially (eg making clear that you're working with a two layer transformer)
- The abstract and intro do not clearly state that you are working with a toy model on a toy task, which I was confused by, and think is somewhat misleading. I recommend editing to clarify, since this is a crucial detail.
- I disagree with your use of superposition to describe the residual stream containing multiple pieces of information. Superposition specifically refers to the model compressing in more meaningful directions than it has dimensions, resulting in non-orthogonal vectors. But in toy models, there could easily be fewer directions than dimensions in an overparametrised model (see eg Progress Measures for Grokking via Mechanistic Interpretability), and so these could be orthogonal. I would just call it a linear representation
- The "information superposition in working spaces" section is, in my opinion, a fairly standard idea in mechanistic interpretability, as discussed as "the residual stream as an information bottleneck" in A Mathematical Framework for Transformer Circuits, which could be helpful to cite.
- The abstract implied that the paper may study the emergence of iteration heads over training ("how CoT reasoning emerges"), which doesn't really happen, and so felt a bit misleading.
- In line 24 the work is motivated as our understanding of why transformers gain CoT abilities is limited. But isn't this clearly because they are trained on examples of people's thought process, eg worked answers to maths problems? More broadly, I think the authors somewhat overstate the degree to which CoT is surprising - isn't it obvious that if you let an LLM break a problem down into small steps, and do each in a separate forward pass, it performs better? The interesting question is how and when this is done.
- Line 40: I found "from which we deduce the usefulness of data curation" confusing on first read, and wasn't sure what was meant (even after reading the whole paper I'm a big confused)
- I thought the Related Work section was fairly sparse (though I understand the pain of conference page limits!). I might flesh it out by discussing how other toy models work explore applicability to LLMs, or comparing to other work that tries interpreting chain of thought.
- Line 64: This was confusing as CoT *is* "next-token prediction". Maybe "Single step next-token prediction" would be clearer?
- Line 202: It is shown that fine-tuning just the final MLP layer enables high accuracy on parity, and claimed that this shows it is computing the successor function. This seems too strong to me - it shows that the final MLP layer is *sufficient*, but not that other parts of the model aren't doing successor too
- Line 341: The comment on "inner circuits" dedicated to multistep reasoning seems a bit of an overclaim to me. This is a plausible claim in general, but doesn't clearly follow from your results, as you only study models trained on single tasks (or trained on one then finetuned on another), while having a single circuit simultaneously work for many tasks is harder

**Limitations:**

See Questions

---

> ### Author Rebuttal · Authors · 2024-08-05
>
> Thank you for your detailed and thoughtful review of our paper. We appreciate your feedback and your many valuable suggestions. To answer your major concerns, we detail our perspective below.
>
> Regarding the relevance of our toy model to real-world LLMs, we agree that our setup is simple and may not capture the full complexity of chain-of-thought reasoning in LLMs. One may complexify iteration heads by designing a first layer that segments tokens into different coherent blocks of varying lengths and outputs positional information $p_t$ at the end of each block, allowing then to iterate over blocks of varying length rather than individual tokens. Regarding real-world experiments, while we appreciate your experimental suggestions, we believe that a serious study requires a different mindset and goals than our paper, which aimed to gain intuition in a controlled setup. With this perspective in mind, we believe that our toy model offers many valuable insights.
>
> Our study offers a plausible story for the usefulness of training on structured data like code: it increases the likelihood of developing clean reasoning circuits, akin to the iteration head, that could be reused beyond code. To this end, we remark that if we blend both the parity and polynomial data, we learn a unique circuit used for both tasks, the `problem` token being recovered at the second layer to condition the MLP implementing the successor function. We did not emphasize these results in our original draft and will revise it. Moreover, the percentage of polynomial data in the blend will modify the training time needed to achieve 100% accuracy on both tasks, hence our remark on "the usefulness of data curation".
>
> Although we have not studied how CoT emergences from a training dynamics perspective, we have a good intuition of the conditions under which the iteration head appears. For example, the parity problem is sometimes solved by recomputing partial sums before spitting out each new output token. This is particularly the case when the sequence lengths are short compared to the capacity of the transformer (which relates to the embedding dimension). On the contrary, when sequences are long, we learn the iteration head, which is a more parsimonious solution. When the sequence length is really long, and the successor function is really simple, the first layer MLP has to learn many position subtractions, while the second MLP layer has not much to learn. This creates excessive pressure on the weights of the first MLP layer, which tends to be alleviated by doing partial pre-computation earlier one, and only learning to perform subtraction for even key positions (Figure 4, right).
>
> We hope that our answer will help you better understand and appreciate our perspective. We also thank you for your minor concerns that we will integrate when reworking our draft.

---

### Official Review · Reviewer_uh9V · 2024-07-13

**Soundness:** 3
**Presentation:** 3
**Contribution:** 3
**Rating:** 7
**Confidence:** 4

**Summary:**

The authors propose a simplified formatting of the input, that represents the structure of chain-of-thought reasoning.  They then propose a theoretical circuit that can efficiently copy information from different indices in the input to the appropriate position in the chain-of-thought to perform an additional iteration step.  The authors create a new metric - attention peakiness - that measures how similar an attention pattern is to one that would have been generated by the theoretical circuit.  They show that several transformer models trained on various iterative tasks in the specified format learn *iteration heads* as measured by the peakiness scores.

The authors follow up with an experiment to show how transfer learning involves repurposing existing circuits for a new task.  The models take longer to learn the parity task than the other tasks in the paper.  However, when you first train a model on a simpler task that involves learning the iteration head, and then finetune it on the parity task, the model can learn parity with roughly an order of magnitude less total compute.

**Strengths:**

- This paper proposes an original copying mechanism that will be of relevance to other researchers studying information flow and circuits in transformers.
- The theoretical circuit for copying information to the correct token positions is well explained.  Figure 1 clearly illustrates how the mechanism works.
- The results from the inductive biases section provide evidence for the generality of iteration heads, showing how they are useful for several different tasks.  This is a novel methodology that could be incorporated into other papers in the field of mechanistic interpretability.
- The authors test for iteration heads in models trained on a variety of tasks (Polynomial Iteration, Parity), on a variety of embedding sizes, and sequence lengths.

**Weaknesses:**

- There wasn't any discussion in the paper regarding how the iteration head relates to previously found copying circuits such as the induction head [1].  Induction heads can also apply "pointer arithmetic on position embeddings" to find token positions to copy information from (see Q-composition in induction heads).  In general, this paper would benefit from more detailed discussion about how this newly discovered mechanism relates to other known internal mechanisms from the mechanistic interpretability literature.
- Chain-of-thought in natural language lacks the structure present in the tasks from the paper.  The model cannot just use "pointer arithmetic" to locate the relevant information in the context, as there is no guarantee on how the input will be structured.  As a result, it is unclear whether actual language models would develop iteration heads to use in chain-of-thought reasoning.
- The primary evidence for trained models learning iteration heads comes from observing attention patterns.  Other works in the field of mechanistic interpretability have moved beyond using attention patterns as definitive evidence for mechanistic theories as they are insufficient to pin down the exact functionality of the model [2].  The paper lacks any additional evidence for the mechanisms beyond analyzing attention scores, such as linear probing [3], or activation patching [4].


[1] Olsson, et al., "In-context Learning and Induction Heads", Transformer Circuits Thread, 2022.

[2] Kaiyue Wen, Yuchen Li, Bingbin Liu, Andrej Risteski. "Transformers are uninterpretable with myopic methods: a case study with bounded Dyck grammars." (2023).

[3] Guillaume Alain, and Yoshua Bengio. "Understanding intermediate layers using linear classifier probes." (2018).

[4] Kevin Wang, Alexandre Variengien, Arthur Conmy, Buck Shlegeris, Jacob Steinhardt. "Interpretability in the Wild: a Circuit for Indirect Object Identification in GPT-2 small." (2022).

**Questions:**

- Including an additional metric for detecting iteration heads in addition to the peakiness score would greatly benefit the paper, such as a linear probe after the first layer trained to predict the index of the EoI token.
- How does the iteration head relate to other copying mechanisms in language models such as induction heads?

**Limitations:**

The authors mention the limits of the toy models and controlled settings in the conclusion.  The authors also mention some of the potential broader impacts of mechanistic interpretability research on society.

---

> ### Author Rebuttal · Authors · 2024-08-05
>
> Thank you for your positive review and valuable feedback on our paper. We appreciate your concerns and will address them below.
>
> Regarding the relationship between iteration heads and other copying mechanisms in language models, such as induction heads, we understand that induction heads were introduced as an example of how to move information across "registers" (the $(e_{t,l})$) with copying mechanisms. The induction head circuit is based on a copy of the previous token computed thanks to position information, together with a copying mechanism based on semantic information. The iteration head relies on another copying mechanism to move information around, in particular, the iteration copy mechanism is purely positional. As for the $Q$-composition you mentioned, some operations in an iteration head could be performed jointly by several matrices (i.e., by "composing" them). For example, the position subtraction could be performed jointly by the first attention value and output matrices, the first layer MLP and the second attention query and key matrices.
>
> As mentioned in the general rebuttal, our goal was not to elicit the appearance or non-appearance of iteration head in production-size LLMs, but come up with controlled experiments to shed lights on LLMs. As a side note, one may generalize iteration circuit to natural language examples beyond pointer arithmetic. For example, one may design a first layer that recognizes coherent semantic information of varying token length, and only associates to the end of these coherent blocks some position information encoded as a vector $p_t$. From there, the circuit would only iterate over these blocks of varying length, allowing to generalize the iteration circuit to natural language examples.
>
> Regarding the use of attention patterns as evidence for mechanistic theories, we remark that the attention operation is the sole one that moves information between registers across time (the $(e_{t,l})$ for varying $t$). As a consequence, the attention maps capture information regarding movement across positions. Moreover, in our setting, linear probing of the first keys will necessarily recover the EoI if the attention is focused on EoI, since the scalar product of the keys with a query could be seen as a linear probe (with weights defined by the query) looking for EoI among the different keys.
>
> Thank you very much for your questions, we will integrate these answers to enhance our manuscript readability and relate it more explicitly with induction heads, and other probing techniques.

---

> > ### Comment · Reviewer_uh9V · 2024-08-12
> > **Reply to Rebuttal**
> >
> > Thank you for the reply.  I believe that adding additional discussion about induction heads will greatly contribute to the paper.
> >
> > > As a consequence, the attention maps capture information regarding movement across positions.
> >
> > My point was that even low attention scores indicate some information movement.  Even if the attention scores were completely uniform, there is still a chance that the model might be doing an "iteration head"-type operation.  This makes it problematic to rely purely on attention patterns as an interpretability tool.
> >
> > I will keep my current score.

---

> ### Author Response · Authors · 2024-08-12
> **Thank you**
>
> Thank you very much for your comment, we better understand your perspective: it is true that we could intervene to "patch" the ideal attention maps and see if this degrades performance, so to check that the final prediction is only based on the route defined in our idealized iteration heads. We will launch this experiment and let you know how it goes.
> Is there anything we could do to help you better appreciate the paper?

---

> > ### Comment · Reviewer_uh9V · 2024-08-12
> > **Reply**
> >
> > Yes, that would be one valuable experiment.  Zero-ing out the attention score between $p_{L+t}$ and the EoI token in the first head, or zero-ing out the attention score between $p_{L+t}$ and $p_{t}$ should bring the model's accuracy to random chance.

---

> > > ### Author Response · Authors · 2024-08-12
> > > **Thank you!**
> > >
> > > Thank you very much. We just ran a few experiments, and zeroing out does mess up the circuit, bringing the performance close to random.
> > > On the contrary, patching the ideal attention maps (i.e. zeroing other routes) does not break the perfect accuracy.
> > > We also realized that patching the wrong attention maps (e.g., focusing on EoI - 2 tokens instead of EoI for the first attention map) leads to funny patterns, whose analysis may lead to an even deeper understanding of the inner mechanisms of the network.
> > > Thank you really much for your suggestion!

---

> > > > ### Comment · Reviewer_uh9V · 2024-08-13
> > > > **Reply**
> > > >
> > > > I appreciate the additional experiment.  As a result, I have raised my score.  I look forward to the inclusion of this experiment in the paper in the final version.

---

> > > > > ### Author Response · Authors · 2024-08-13
> > > > >
> > > > > Thank you very much for raising your score, and thank you even more for providing us valuable insights

---

### Official Review · Reviewer_YdTm · 2024-07-13

**Soundness:** 3
**Presentation:** 3
**Contribution:** 2
**Rating:** 7
**Confidence:** 4

**Summary:**

This paper provides a mechanistic study of small transformers trained to perform simple algorithmic tasks (polynomial iteration, parity, binary copy) with scratchpads. Findings show that trained transformers exhibit attention patterns dubbed "iteration heads": heads that iterate through the input, performing index arithmetic to determine which position from the original input to process in the next scratchpad step. The paper also provides a theoretical circuit construction performing this. A variety of experiments are provided to examine the behavior.

**Strengths:**

* The paper addresses a timely question, with little prior work on the same question.
* The paper links the outcome of probing with an intuitive explicit construction.
* The paper includes convincing analysis of the learned circuit, with ablation and evidence of skill transfer.

**Weaknesses:**

* I believe the biggest weakness is an overclaim issue: While the title and the abstract frame the paper in terms of "chain-of-thought", the setup considered is in fact that of *supervised* training with scratchpads [2], not that of classical chain-of-thought prompting where the ability to do CoT emerges from general pretraining [1]. The paper provides no results from CoT in language models. While the paper claims implications for larger LLMs (line 42-48), there is very little support for these claims. I believe the term "scratchpad" would be more appropriate than "chain-of-thought" for the subject matter of the paper.
* There is no theoretical understanding about why this particular circuit would come out of SGD training.  Learning theory is admittedly hard, but ths should be acknowledged clearly as a limitation.
* Empirically, an important benefit of scratchpads lies in improved length generalization [3], whereas the setup here tests only on in-domain lengths. It would be interesting to know what happens when evaluating at longer lengths.

[1] Wei et al, 2022, Chain-of-Thought Prompting Elicits Reasoning in Large Language Models, https://arxiv.org/abs/2201.11903 [cited in the paper]

[2] Nye et al, 2021, Show Your Work: Scratchpads for Intermediate Computation with Language Models, https://arxiv.org/abs/2112.00114 [cited in the paper]

[3] Anil et al, 2022, Exploring Length Generalization in Large Language Models, https://arxiv.org/abs/2207.04901

**Questions:**

Related findings about mechanistic analysis of CoT-like computation (only with 2 steps, but in a setting where it actually emerges from general next-token prediction) is provided in [1] (particularly their Figure 14). The paper under review here might benefit from clearly pointing out how it provides novelty over that result (e.g., in that it considers longer chains and more extensive analysis).

[1] Hahn and Goyal, 2023, A Theory of Emergent In-Context Learning as Implicit Structure Induction, https://arxiv.org/abs/2303.07971

**Limitations:**

The paper should be more upfront about some limitations, as described under "Weaknesses".

---

> ### Author Rebuttal · Authors · 2024-08-05
>
> Thank you for your positive review and valuable feedback on our paper. We appreciate your concerns and will address them below.
>
> We acknowledge that there is some ambiguity between the concepts of "scratchpad" and "chain-of-thought" and will review our manuscript to make it clearer. In our view, scratchpads reasoning is a generic method where a LLM edits a scratchpad before giving an answer. In contrast, CoT consists in asking the model to output intermediate steps of reasoning, as per [1], before providing a final answer. The original CoT paper is a bit confusing as it considers a few-shot learning setting (which relates to in-context learning). Thank you for bringing this to our attention.
>
> Studying the training dynamics of iteration heads seems feasible but not trivial, as the dynamics depend on many details that modify the dynamics (Adam, full batch, architecture, data...) and could be hard to quantify precisely. This is an exciting avenue for future work, especially if it sheds light on the importance of data curation, the possibility of "sharded" circuits, or the difficulty to choose between circuits for the parity dataset.
>
> Regarding length generalization, in our setting, the first layer MLP has to align superposition of embeddings to perform a subtraction. For a position $t$ never seen before, $p_t$ would be a random embedding, and there is no way to generalize. However, if the lengths are sampled with a Zipf law, a transformer may learn to perform well for long sentences, even if these were sparse in the dataset (to generalize, it only has to learn position subtraction, which can be done in a sample-efficient manner).
>
> Thank you very much for the pointer to [2]. This paper is quite interesting, taking an information-theory and minimum description length perspective to shed light on ICL, CoT, composition in LLMs. In contrast, we focus on iteration heads as one "reasoning template" that could be learned by LLMs, focusing on its emergence and transferability when varying the training data. It allows us to perform more targeted ablation studies and give more concrete insights, for example, we also observe "abrupt growth of training and testing losses", but no "grokking" of the position subtraction (that stays in a "memorization" regime).
>
> [1] Kojima et al., 2023. Large Language Models are Zero-Shot Reasoners, [https://arxiv.org/abs/2205.11916](https://arxiv.org/abs/2205.11916)
> [2] Hahn and Goyal, 2023, A Theory of Emergent In-Context Learning as Implicit Structure Induction, [https://arxiv.org/abs/2303.07971](https://arxiv.org/abs/2303.07971)
>
> We hope this clarifies our position and addresses your concerns. Thank you again for your valuable feedback.

---

> > ### Comment · Reviewer_YdTm · 2024-08-12
> >
> > Thanks for the response. Expecting that the authors will incorporate the suggestions to address the weaknesses stated, I am raising my score.

---

> ### Author Response · Authors · 2024-08-12
> **Thank you**
>
> Thank you very much for raising your score, and your valuable comments that will help us improve our draft.
> Best regards

---

### Author Rebuttal · Authors · 2024-08-05

We appreciate the time and effort the reviewers spent reviewing our paper. The overall reception has been positive. The reviewers appreciate the "iteration head" definition and experiments regarding its transferability across tasks.

Reviewers expressed concerns that will help us greatly improve our work. In particular, we should have been clearer that the goal of our paper is not to study iterative circuits in production-size LLMs (indeed, ChatGPT was not able to solve our parity problem at writing time), but to introduce a controlled setting that enables the rigorous study of phenomena that are relevant for LLMs.
Following the reviewers suggestions, we will add more references to related works, and compare iteration heads with induction heads more thoroughly.

In order to help reviewers better appreciate our perspective, let us emphasize some results we appreciate from our study:
* Iteration heads illustrate how transformers learn generic reasoning patterns that could be used at inference time. Moreover, it showcases how they may learn to generate many tokens before providing a final answer through teacher-forcing. We expect similar (yet more complex) phenomena to explain the emergence of CoT in larger scale models.
* We showcase how the apparition of these circuits depends on the training data mix (see Figures 3, 5, and 6).
* Iteration heads appear frequently in our practical settings. However, in extreme situations, we observe the emergence of alternative circuits: for example, Figure 9 shows that when using three layers, the parsimonious circuit can be "sharded" across layers, making the transformer more difficult to comprehend if we were not aware of iteration heads.


An avenue we are excited to explore in future work is studying how to "un-shard" circuits (i.e. going from the attention maps of Figure 4 or 9 to the ones of Figure 2). We believe that our controlled setup provides a good testbed toward architectural and optimizer changes in order to ensure that semantically coherent operations (e.g. the position subtraction, or the successor function) are performed in single layers.

While we hope that our generic answer will help you better understand and appreciate our perspective, we will stay at the reviewer's disposal during the discussion period to clear any remaining concerns.

---

### Author Response · Authors · 2024-08-13
**Extended thanks**

As the discussion period comes to a close, we would like to extend our sincere appreciation to all of the reviewers for their time and effort. We were thoroughly impressed by the high quality of the reviews and the valuable suggestions provided to enhance our work.
It is not uncommon to hear complaints about the quality of reviews and the low level of commitment to serving the community, but we want to emphasize that this does not at all reflect our experience.
The dedication and constructive feedback provided by the reviewers were truly exceptional.
Thank you very much to each and every one of you.

---

### Decision · Program_Chairs · 2024-09-25

**Decision:**

Accept (poster)

**Comment:**

This paper studies how small transformers are trained to perform simple algorithmic tasks (polynomial iteration, parity, binary copy) with scratch pads. The authors demonstrate that transformers exhibit iteration heads, which iterate through the input and perform index arithmetic to determine which position of the original input to process. The authors also conduct some experiments to verify their findings and show empirically that simple two-layer transformers can learn to implement the iteration-head algorithm when trained on iterative tasks.

The reviewers found the paper to be technically sound and impactful, with a recommendation for acceptance after addressing some weaknesses and concerns. This work will contribute to the understanding of the mechanism behind transformers performing algorithmic tasks.